# "Know your epidemic, know your response": Epidemiological assessment of the substance use disorder crisis in the United States

**Andrés Hernández** [1,2,3], **Minxuan Lan** [2], **Neil J. MacKinnon** [4,5], **Adam J. Branscum** [6], **Diego F. Cuadros** [1,2,5] *

**1** Health Geography and Disease Modeling Laboratory, University of Cincinnati, Cincinnati, Ohio, United States of America, **2** Department of Geography and GIS, University of Cincinnati, Cincinnati, Ohio, United States of America, **3** Department of Geo-information Processing, Faculty of Geo-Information Science and Earth Observation, University of Twente, Enschede, The Netherlands, **4** Department of Population Health Sciences, Medical College of Georgia, Augusta, Georgia, United States of America, **5** Geospatial Health Advising Group, University of Cincinnati, Cincinnati, Ohio, United States of America, **6** Department of Biostatistics, College of Public Health and Human Sciences, Oregon State University, Corvallis, Oregon, United States of America

* diego.cuadros@uc.edu

**Data Availability Statement:** The data underlying the results presented in the study are available from the U.S. Center for Disease Control and

## Abstract

The United States (U.S.) is currently experiencing a substance use disorders (SUD) crisis with an unprecedented magnitude. The objective of this study was to recognize and characterize the most vulnerable populations at high risk of SUD mortality in the U.S., and to identify the locations where these vulnerable population are located. We obtained the most recent available mortality data for the U.S. population aged 15–84 (2005–2017) from the Centers for Diseases and Prevention (CDC). Our analysis focused on the unintentional substance poisoning to estimate SUD mortality. We computed health-related comorbidities and socioeconomic association with the SUD distribution. We identified the most affected populations and conducted a geographical clustering analysis to identify places with increased concentration of SUD related deaths. From 2005–2017, 463,717 SUD-related deaths occurred in the United States. White population was identified with the highest SUD death proportions. However, there was a surge of the SUD epidemic in the Black male population, with a sharp increase in the SUD-related death rate since 2014. We also found that an additional average day of mental distress might increase the relative risk of SUD-related mortality by 39%. The geographical distribution of the epidemic showed clustering in the West and Mid-west regions of the U.S. In conclusion, we found that the SUD epidemic in the U.S. is characterized by the emergence of several micro-epidemics of different intensities across demographic groups and locations within the country. The comprehensive description of the epidemic presented in this study could assist in the design and implementation of targeted policy interventions for addiction mitigation campaigns.

Prevention (CDC) vital statistics (https://www.cdc.
gov/nchs/nvss/index.htm).

**Funding:** The authors received no specific funding
for this work.

**Competing interests:** The authors have declared
that no competing interests exist.

## Introduction

Substance use disorders (SUD) have been declared one of the top public health priorities in the
United States (U.S.), with 185 SUD-related deaths, on average, each day in 2018 [1, 2]. SUD
disorders are considered a subgroup of the addiction diseases that are deemed as mental health
conditions in which a person repeatedly uses substances or engages in behaviours with the
knowledge of their harmful consequences [3]. In the U.S., it is estimated that one in five people
aged 12 years or older used an illicit drug, and 8.1 million had an illegal drug use disorder in
2018 [4], with 67,367 reported deaths by drug overdose in the same year [1, 2]. Overall, the U.
S. mortality rate related to SUD reached 20.7 deaths per 100,000 inhabitants in 2018, with
West Virginia (51.5), Delaware (43.8), Maryland (37.2), Pennsylvania (36.1), Ohio (35.9), and
New Hampshire (35.8), having the highest mortality rates at the state level [2].

Several studies have examined multiple characteristics of the addiction epidemic in the U.S.
These studies have reported a significant increase in mortality rates from 2010, with its highest
peak in 2017, and a considerable demographic and spatial heterogeneity of the epidemic being
attributed, in part, to the uneven distribution of several demographic and socioeconomic fac-
tors and health comorbidities across the country [5–7]. However, previous studies have not
fully explained the reasons behind the unequal spread of the SUD epidemic and there remains
a need to reduce the high level of SUD-related mortality rates in the country. As a result, sev-
eral sociological studies have suggested the need for implementing a socio-ecological frame-
work to conceptualize the drivers of addictive behaviours according to their level of influence
in order to design effective strategies [8, 9]. These studies highlight the importance of the inter-
connection between individual and broader social and environmental domains as essential to
understanding the SUD epidemic. Within this framework, individual, family, neighborhood,
and community-level attributes have been identified as potential drivers of the current SUD
epidemic [7–9]. Furthermore, our preliminary study conducted in Ohio identified different
spatial and demographic distributions associated with the opioid overdose deaths in the state,
such that the epidemic is concentrated in specific demographic groups and locations, with
multiple spatial and temporal sub-epidemics emerging at distinct time periods [10].

Successful approaches like the "Know your epidemic, know your response" framework
implemented for countering the malaria and HIV epidemics worldwide have resulted in
mitigation policies that shifted from intervention strategies (i.e. vaccines, medical treatment)
to targeted prevention plans (i.e. modifying behavioral response of individuals) [11]. The core
of the "Know your epidemic, know your response" approach is the identification of the envi-
ronmental, socioeconomic, and demographic drivers of an epidemic [6, 12, 13]. These drivers
become the cornerstones of the design and implementation of prevention measures that target
vulnerable populations under their unique social, environmental, and epidemiological circum-
stances [11]. Moreover, "Know your epidemics, know your response" approach highlights the
role of the individual awareness of the risk in the ability to respond with appropriate mitiga-
tion strategies, allowing to focus on education efforts and mitigation of risk factors, more than
in allocating resources for intervention policies [14]. Similar to malaria and HIV, addiction
disorders are characterized by complex spatial hierarchical structures caused by multiple con-
current sub-epidemics of different intensities among different populations [11]. However, in
the case of SUD-related mortality rates, the link between community-level factors and risk of
death is not well understood. In addition, the vulnerable populations suffering the highest bur-
den of the SUD epidemic driven by specific socioeconomic characteristics and comorbidities
are still not well characterized. Epidemiologic research to resolve these complexities should
address the spatial and hierarchical nature of the epidemic to estimate associations between
individual- and community-level attributes and SUD-related mortality.

Against this background, we used data from the U.S. Centers for Disease Control and Prevention (CDC) on individual mortality from 2005–2017 to analyze the demographical, spatial, and temporal structure of the SUD epidemic and its associated risk factors in the U.S. In accordance with the "Know your epidemic, know your response" approach, the aim of this study is two-fold: (i) to identify and characterize the demographic groups at highest risk of death by SUD, and (ii) to describe the spatial and temporal dynamics of the SUD epidemic in the U.S. We aimed to identify the key demographic factors associated with the epidemic, and the vulnerable populations and places where the burden of the epidemic is concentrated. A nationwide description of the epidemic would assist in the design and implementation of targeted policy interventions for addiction mitigation campaigns through an understanding of the spatial variability and epidemiological profiles in the U.S.

## Research methods

### Data sources description, sampling, and demographic analysis

Data were provided by the CDC from restricted-use vital statistics micro-data files for the period of January 2005 to December 2017, which is the latest available mortality data at the time of the analysis [15]. Available data included the date and county of death, demographic characteristics of individuals (sex, race, age, marital status, and educational level) and the International Classification of Diseases, 10th Revision (ICD-10) code for the cause of death [16]. We extracted information about drug overdose deaths for individuals aged 5 to 84 years from ICD-10 codes for unintentional substance poisoning. Monthly death rates by county were computed as the ratio of the number of SUD deaths to the number of total deaths and were scaled by 1,000.

Community-level factors related to health behaviours and physical and mental health at the county level were retrieved from the County Health Rankings & Roadmaps program from 2010 to 2017 [17]. These covariates corresponded to social and health risk factors that have been associated with SUD in previous studies at the community level [9, 18, 19]. We included the self-reported number of days per month under physical and mental distress, excessive adult drinking, and tobacco consumption from the Behavioral Risk Factor Surveillance System (BRFSS) [20]. We also included the percent of children living in poverty and the population without health insurance in each county as potential socioeconomic factors associated with the SUD epidemic.

In addition, from the complete data set provided by the CDC, we performed stratified random sampling with strata given by year and state of death occurrence to avoid requiring excessive computational resources for regression analysis [21, 22]. Finally, SUD death rates by demographic groups were visualized using time series graphs and heat maps to describe the temporal dynamics of the SUD epidemic from 2005 to 2017. We computed death rates by race, gender, and age group to determine the groups most affected by the epidemic. Demographic analysis was conducted using the complete data and also data from the stratified random sampling. Institutional Review Board Approval was not necessary for this study because all data were deidentified and publicly available.

### Risk factors associated with mortality caused by substance use disorders

We conducted logistic regression analyses of data collected from stratified random sampling to identify individual- and community-level factors associated with the odds of SUD-related mortality. The binary outcome variable for each study subject was death by SUD (y = 1) or death by other causes (y = 0). Individual-level covariates were age group (by quinquennial), race (White, Black, other), sex (female, male), educational level (primary, secondary, college or

higher), and marital status (never married, currently married, and previously married). The logistic regression model was implemented using a mixed effects generalized additive model [23] (GAM) that allowed for nonlinear trends for all of the community-level covariates (individual-level covariates are all categorical) [24]. Our primary analysis used a logistic regression GAM mixed model for evaluating associations between individual- and community-level covariates and SUD-related mortality without including interaction terms. A supplementary analysis added interaction terms between individual- and community-level covariates (mental and physical health) to the model. All logistic regression models included a random effect for county. All sampling operations were conducted using Python 3.8 [25], and Spark 4.1 [26] with the pyspark package, and statistical analyses were conducted using R version 3.5.2 (R Project for Statistical Computing) [27] with the mgcv 1.8–31 package [28].

### Cluster analysis and spatiotemporal risk estimation

Spatial clusters of SUD-related deaths were identified using scan statistics implemented in the SaTScan software [29]. Locations in the U.S. where the number of deaths due to SUD was higher than expected under the null hypothesis of a homogeneous distribution of SUD related deaths were classified as hotspots. The number of SUD-related deaths from the complete dataset at the county level from 2005 to 2017 were analyzed using a Poisson model with the total number of deaths from any cause by county included as an offset. Resulting hotspots were selected based on having p-values less than 0.05 and filtered to contain at least three counties and non-overlapping clusters. Community-level covariates were computed for each hotspot, all hotspots combined, and non-hotspot areas.

In addition, we assessed the spatial and spatiotemporal dynamics of the relative risk (RR) of SUD-related mortality using a Bayesian zero-inflated Poisson regression model to accommodate excess zero counts in sparse area data in the context of a Besag-York-Mollie (BYM) model [30]. The spatial analysis was computed by counties within the contiguous U.S. with available community-level information and was applied to the total number of deaths from 2005 to 2017, while the spatiotemporal study used the deaths by county, aggregated by semester from 2005 to 2017. The model was fitted using an integrated nested Laplace approximation implemented in the R-INLA software package [31]. Results of these analyses were mapped using the R statistical software along with the ggplot2 [32] library for spatial visualization. Extended details of the methods can be found in the S1 Text.

## Results

### General demographic profile of the SUD epidemic in the U.S.

Table 1 presents the distribution of deaths caused by SUD in the selected demographic groups, with 463,717 SUD-related deaths (2.04%) among the total number of deaths (22,705,614) registered in the U.S. from 2005 to 2017. Males had a higher proportion of SUD-related deaths (2.38%) compared to females (1.61%) in all racial groups. Additionally, the proportion (2.14%) of SUD-related deaths for the White population was higher than that for the Black population (1.60%), and other races (1.37%). Fig 1 illustrates the temporal trajectories of SUD-related death rates per 1,000 total deaths by race and sex, with the White male population consistently having the highest SUD-related mortality rates from 2005 to 2017. However, SUD-related death rates for Black males have increased sharply since 2014, going from 18.91 (2014–01) to 38.65 (2017–12) with a percentage change (PC) of 104,38%, compared to the White males increase from 26.46 (2014–01) to 39.77 (2017–12), PC: 50.30%. In addition, the heat maps in Fig 2A and 2B show the temporal patterns of SUD death rates by race, sex, and age groups, and indicate a concentration of SUD-related deaths among individuals aged 15 to 39 in both

**Table 1. Number of deaths in the United States caused by Substance Use Disorders (SUD) and all other causes from 2005 to 2017.**

| Risk Factor | Deaths by SUD | Deaths by any other cause | Proportion of SUD Deaths |
|---|---|---|---|
| **Age Group** | | | |
| < 15 | 377 | 73,303 | 0.51% |
| 15–19 | 8,347 | 139,728 | 5.64% |
| 20–24 | 34,544 | 226,495 | 13.23% |
| 25–29 | 50,657 | 237,945 | 17.55% |
| 30–34 | 54,039 | 265,932 | 16.89% |
| 35–39 | 53,456 | 345,450 | 13.40% |
| 40–44 | 56,154 | 528,421 | 9.61% |
| 45–49 | 64,612 | 870,268 | 6.91% |
| 50–54 | 61,052 | 1,357,145 | 4.30% |
| 55–59 | 43,568 | 1,854,540 | 2.30% |
| 60–64 | 21,220 | 2,282,678 | 0.92% |
| > 64 | 15,570 | 14,046,878 | 0.11% |
| Not Available | 121 | 13,114 | 0.91% |
| **Sex** | | | |
| Females | 159,520 | 9,754,240 | 1.61% |
| Males | 304,197 | 12,487,657 | 2.38% |
| **Race** | | | |
| White | 404,088 | 18,483,493 | 2.14% |
| Black | 50,111 | 3,074,175 | 1.60% |
| Other | 9,518 | 684,229 | 1.37% |
| **Educational Level** | | | |
| Primary | 7,482 | 874,752 | 0.85% |
| Secondary | 57,420 | 2,940,637 | 1.92% |
| College-Level | 23,051 | 1,550,141 | 1.47% |
| Not Available | 375,764 | 16,876,367 | 2.18% |
| **Marital Status** | | | |
| Never Married | 207,904 | 3,164,284 | 6.17% |
| Currently Married | 111,295 | 10,010,622 | 1.10% |
| Previously Married | 134,779 | 8,800,899 | 1.51% |
| Not Available | 9,739 | 266,092 | 3.53% |
| **Total** | **463,717** | **22,241,897** | **2.04%** |

sexes and all race groups, with an additional clustering of deaths in Black males aged 40–49. Fig 2A illustrates the SUD-related mortality rates peaking for white population during the first semester of 2017 with the highest rates on White young males (350 SUD-deaths per 1,000 total deaths), in contrast to the Black young males (Fig 2B) with 140 SUD-related deaths per 1,000 total deaths. The substance discrimination analysis, which identified different substances leading the epidemic in different populations, is included in the S1 Fig.

## Socioeconomic factors and comorbidities associated with the SUD epidemic

Results from the multilevel mixed effect logistic regression GAM model over the stratified sample are presented in Table 2 for the individual covariates and in Fig 3 for the county-level variables. The statistical characteristics of the stratified sample are described in S1 Table. Five percent of the total number of registered deaths in the U.S. from 2005 to 2017 were included

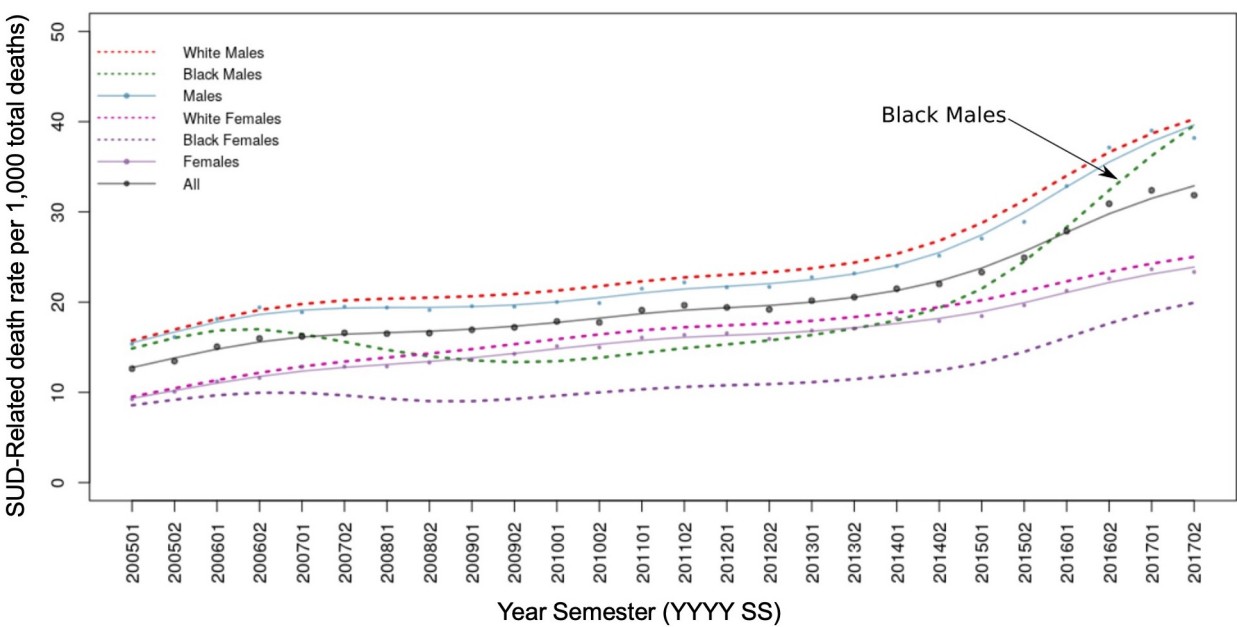

**Fig 1. Descriptive demographics of the Substance Use Disorder (SUD) death rates (SUD deaths / total deaths \* 1,000) per semester by major demographic groups in the U.S. (2005–2017).**

in the sample (1,111,199 deaths, with 22,483 or 2.02% prevalence of SUD-related deaths). We found that age, race, educational level, and marital status were significantly associated with the odds of death by SUD. Individuals aged 25–29 years had the highest odds of SUD-related mortality (odds ratio [OR]: 3.71, 95% confidence interval [CI]: 3.31–4.16) compared to individuals aged 15–19 years, followed by the 30–34 year old age group (OR: 3.65, 95% CI: 3.26–4.09) and 20–24 year old (OR: 2.58, 95% CI 2.30–2.91). Whites had more than double the odds of death

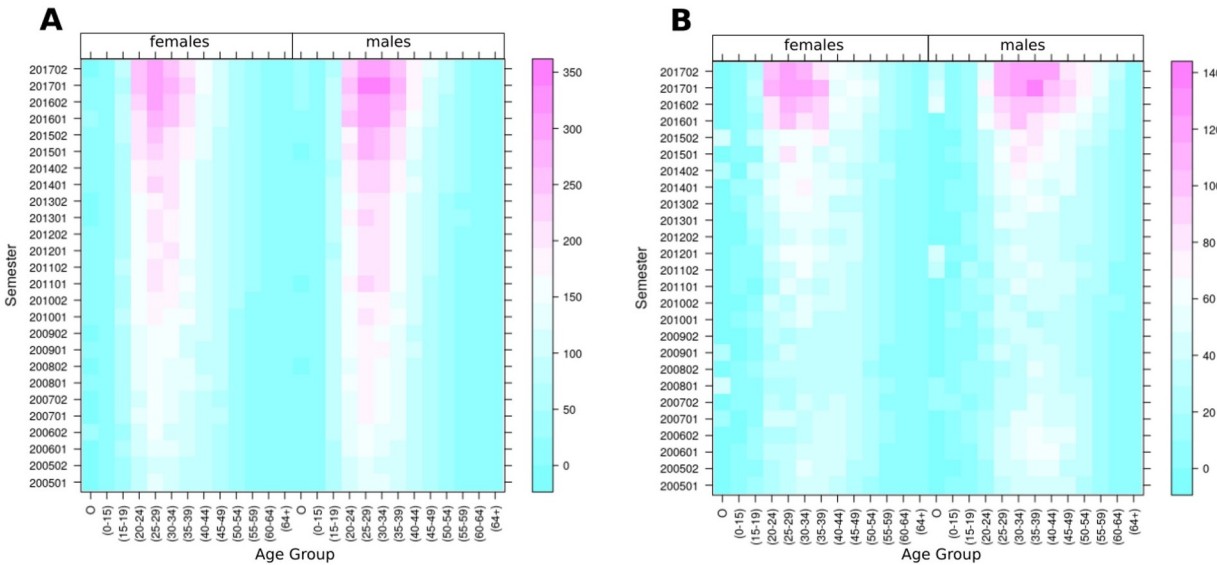

**Fig 2. Substance Use Disorders (SUD) death rates per semester by age groups (SUD-related death rates per 1,000 deaths) (A) for the White population (B) for the Black population.**

**Table 2. Odds ratios for the association of demographic factors with death caused by Substance Use Disorders (SUD).**

| Risk Factor | Odds Ratio | CI 0.025 | CI 0.975 | P-value |
|---|---|---|---|---|
| **Age Group** | | | | |
| < 15 | 0.11 | 0.07 | 0.17 | <0.001 |
| 15–19 | Ref. | | | |
| 20–24 | 2.58 | 2.30 | 2.91 | <0.001 |
| 25–29 | 3.71 | 3.31 | 4.16 | <0.001 |
| 30–34 | 3.65 | 3.26 | 4.09 | <0.001 |
| 35–39 | 2.80 | 2.49 | 3.14 | <0.001 |
| 40–44 | 1.89 | 1.68 | 2.11 | <0.001 |
| 45–49 | 1.32 | 1.18 | 1.48 | <0.001 |
| 50–54 | 0.77 | 0.68 | 0.86 | <0.001 |
| 55–59 | 0.39 | 0.35 | 0.44 | <0.001 |
| 60–64 | 0.15 | 0.13 | 0.17 | <0.001 |
| > 64 | 0.01 | 0.01 | 0.02 | <0.001 |
| Not Available | 0.18 | 0.08 | 0.40 | <0.001 |
| **Sex** | | | | |
| Females | Ref. | | | |
| Males | 1.00 | 0.97 | 1.03 | 0.933 |
| **Race** | | | | |
| White | Ref. | | | |
| Black | 0.45 | 0.43 | 0.47 | <0.001 |
| Other | 0.45 | 0.41 | 0.50 | <0.001 |
| **Educational Level** | | | | |
| Primary | Ref. | | | |
| Secondary | 1.24 | 1.10 | 1.39 | <0.001 |
| College-Level | 0.99 | 0.87 | 1.13 | 0.932 |
| Not Available | 1.82 | 1.62 | 2.04 | <0.001 |
| **Marital Status** | | | | |
| Never Married | Ref. | | | |
| Currently Married | 0.59 | 0.57 | 0.62 | <0.001 |
| Previously Married | 1.19 | 1.14 | 1.24 | <0.001 |
| Not Available | 1.39 | 1.25 | 1.55 | <0.001 |

All odds ratios estimated using logistic regression generalized additive mixed models.

by SUD compared to Blacks (OR: 0.45, 95% CI: 0.43–0.47) and other races (OR: 0.45, 95% CI 0.43–0.47). Those with a secondary level education had higher odds of death by SUD (OR: 1.24, 95% CI: 1.10–1.39) compared to those with a primary education. Married individuals had lower odds of SUD-related death than singles (OR: 0.59, 95% CI: 0.57–0.62) and divorced/ widowed individuals (OR: 1.19, 95% CI: 0.57–0.62). There was no statistical evidence for a difference in the population odds of SUD-related death for males and females.

The same logistic regression GAM analysis indicated that average number of mentally and physically unhealthy days, percentage of children living in poverty, and percentage of the uninsured population were community-level factors associated with the odds of SUD-related death (S2 Table). The average number of mentally and physically unhealthy days were directly (i.e., positively) associated with an increasing the odds of SUD deaths in individuals living in

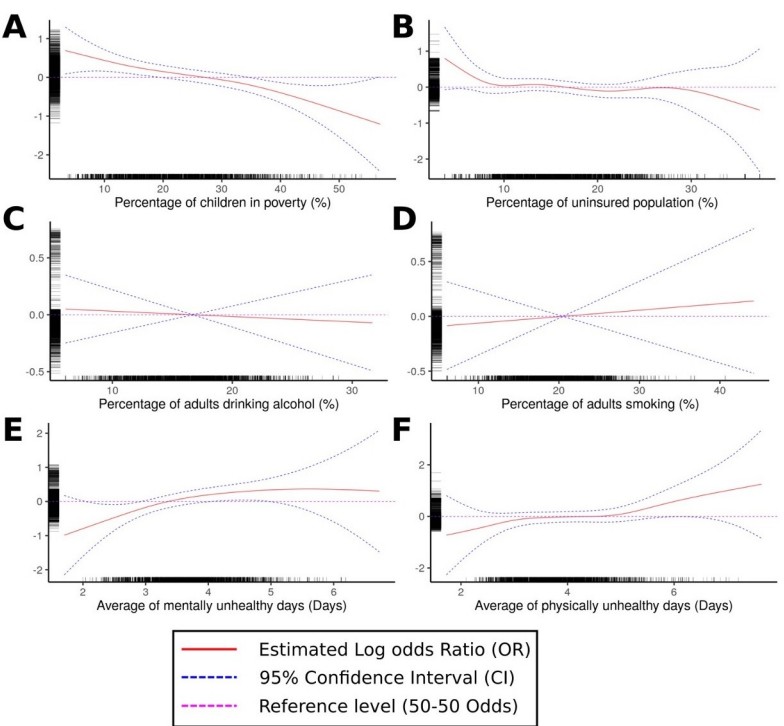

**Fig 3. Generalized additive models estimation for the log odds ratio of death by Substance Use Disorders (SUD) associated with: (A) percentage of children living in poverty, (B) percentage of uninsured population, (C) percentage of the adult population with excessive alcohol consumption, (D) percentage of the adult population that consume tobacco, (E) average of mentally unhealthy days, and (F) average of physically unhealthy days.**

counties with an average of more than 4.0 of mentally and 4.5 of physically unhealthy days (Fig 3E and 3F, respectively). Children living in poverty and uninsured population percentages showed an inverse relationship, with decreased odds of SUD-related deaths in counties with a percentage population of more than 25% (children living in poverty) and 15% (uninsured population). Lastly, the effects of the average number of mentally and physically unhealthy days on each age group, sex, and race included in our supplement showed dissimilar effects of mentally and physically unhealthy days across demographic groups, especially in the age-group interaction model (S2 Table).

## Clustering analysis and spatio-temporal risk estimation

We identified 25 clusters (hotspots) with a significant concentration of SUD-related deaths at the national level from 2005–2017 (S3 Table). The hotspots contained 165,682 (35.73%) of the total reported SUD-related deaths in the U.S., but only included 527 (17.00%) of the 3,111 counties in our clustering and risk estimation analysis. The RR of the hotspots was 1.35 (observed vs. expected SUD deaths) and 2.76% (165,682/5,999,443) of SUD-related deaths relative to all deaths, in comparison to an RR of 0.87 and 1.78% (298,035/16,706,171) in the areas outside the hotspots. Fig 4A shows the location of the SUD-related mortality hotspots, and the spatial distribution of the RR of death by SUD. The estimated RR ranged from 0 to 5.6 and was classified as lowest risk areas (RR < 0.60), low risk (RR: 0.60–1.0), intermediate-risk (RR: 1.00–1.50), high risk (RR: 1.50–2.50), and highest risk (RR> 2.50).

The highest density of SUD-related death hotspots was located on the border areas of the East North Central, Middle Atlantic, East South Central, and South Atlantic regions, including

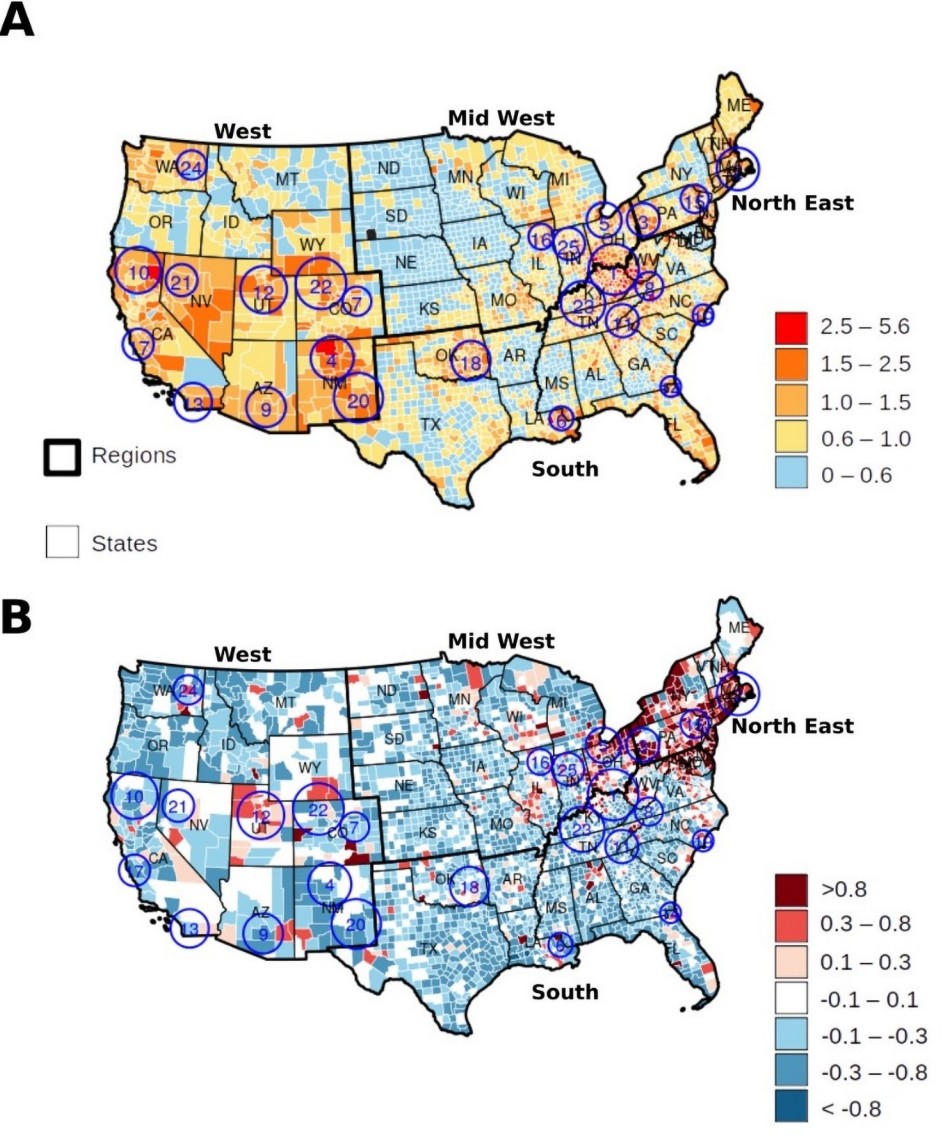

**Fig 4.** (A) Spatial distribution of relative risk for death by Substance Use Disorder (SUD) in the contiguous USA (2005–2017) with identified clusters with (enumerated blue circles). (B) Change of the relative risk (first semester 2005 compared to last semester 2017) with identified clusters of substance overdose related deaths (enumerated blue circles).

in the states of Ohio, Pennsylvania, Kentucky, West Virginia, Indiana, and Tennessee (Clusters 1, 3, 5, 8, 23, 25). From 2005 to 2017, 66,227 SUD-related deaths (14.28% of all SUD-related deaths) occurred in these hotspots (RR = 1.44). Additionally, a second area of high relative risk was found in the Pacific and Southwest, including in the states of California, Utah, Colorado, Arizona, Nevada, and New Mexico (Clusters 4, 7, 9, 10, 12, 13, 17, 20, 21, 22) with 38,348 SUD-related deaths (8.26%) and an average RR of 1.49. Finally, areas with the lowest RR were identified in the West North Central regions, with no clusters, and only 21,875 (4.71%) SUD-related deaths in these areas. The average RR in these areas was 0.50 for North Dakota, South Dakota, Nebraska, Iowa, Kansas, Missouri, and Minnesota. Fig 4B describes the temporal trends of SUD-related mortality risk by estimating the percent change between the RR for the

first semester of 2005 and the RR for the last semester of 2017. Additional estimates of the RR of SUD-related mortality from the Bayesian Poisson spatial regression analysis are summarized in Table 3 and a detailed description included in the results supplement.

## Discussion

We found substantial spatial and demographical variation of the SUD epidemic in the U.S. from 2005 to 2017, which was characterized by the emergence of several micro-epidemics of different intensities across demographic groups and locations within the country. We found that the White male population was the group experiencing the highest rates of SUD-related deaths during this timeframe, and according to our results, 33.82% of the total deaths in White males aged 30 to 34 were caused by unintentional drug-related poisoning during the first semester of 2017. The most vulnerable age-groups among White males were 25–29 (31.34% deaths by SUD), and 30–34 (30.71%) in the second semester 2017, which is the most updated data available in our analysis. However, although the White male population was suffering the highest burden of the epidemic during the study period, a striking surge of the epidemic emerged in the Black male population, particularly in ages 30–34 (12.01%), 35–39 (11.88%), 40–49 (11.59%), and 25–29 (11.37%) by the second semester, 2017.

The demographic disparities identified in this study could be the result of a complex system of sub-epidemics fueled by different substances targeting specific demographic groups, and leading different phases of the epidemic [10, 33]. According to our results, the latest stage of the epidemic has been led by prescription opioids, and, since 2013, by synthetic opioids. Early in the epidemic, Black males were one of the most affected populations, impacted by crack-cocaine substances that were fueling this first wave of the SUD epidemic (during early 1990s), but the rapid increasing in the prescribing of opioids in the following phases of the epidemic boosted the SUD-related death mortality in the White population [34]. However, the increased availability of illegal synthetic opioids and heroin has shifted again the epidemic towards the Black population, with an increase in SUD-related Black males' deaths, particularly in Black males age 45 to 55, who have become one of the most vulnerable populations in the past few years [7].

Additionally, mental and physical distress were found to be key community-level drivers of the SUD epidemic in the country. We found that an additional average day of mental distress might increase the RR of SUD-related mortality by 39% at the county level. Mental health and SUD comorbidity are known as co-occurring disorder or dual diagnosis is a long-known associated illness [35–37]. Managing mental illness in SUD patients can be a key factor in the addiction mitigation, due to a higher probability of addiction relapsing in individuals with

**Table 3. Bayesian Poisson regression spatial analysis of the associations between county-level covariates and the relative risk (RR) of death by Substance Use Disorders (SUD).**

|  | RR | CrI 0.25 | CrI 0.975 |
|---|---|---|---|
| (Intercept) | 0.11 | 0.08 | 0.14 |
| Percentage of children living in poverty | 0.96 | 0.96 | 0.96 |
| Percentage of the population which does not have health insurance (uninsured) | 1.01 | 1.00 | 1.01 |
| Percentage of adults with excessive alcohol consumption | 1.00 | 0.99 | 1.01 |
| Percentage of adults consuming tobacco | 1.01 | 1.00 | 1.01 |
| Average number of mentally unhealthy days | 1.39 | 1.32 | 1.48 |
| Average number of physically unhealthy days | 1.28 | 1.21 | 1.35 |

Values are posterior means of RR, posterior standard deviations (SD) of RR, and 95% credible intervals (CrI) for RR.

mental disorders [38]. Moreover, our results suggest mental distress impacted young adults more commonly in locations where the average mentally unhealthy days exceeds 4.02. Furthermore, we found that an additional average day of physical distress might increase the RR of SUD-related mortality by 28%, and this factor was affecting more older adults with a more pronounced effect in the White population. Characteristics of the spatial distribution of physical distress suggest higher levels in the South and Midwest regions of the U.S., potentially associated with a high prevalence of chronic health conditions, smoking, obesity and physical inactivity, especially higher in women and populations with low SES characteristics [39]. These findings have been previously discussed by other researchers. In particular, Case and Deaton's "*Mortality and Morbidity of 21st Century*" work included a wider examination of mortality rates of midlife population of the U.S. from 1999–2015 [40]. Among their findings, they reported an increase of death rates due to alcohol, suicide, and overdose related causes and their link with an increase of the physical and mental morbidity on the White population [40]. Our study differs from that of Case and Deaton because we focused only on unintentional drug overdoses in a wider age-groups, which potentially limits the scope of age, income and education role on the SUD-related death risk. However, their study also highlights the role of marital status, and revealed a non-clear association of gender and wealth to the increase of the death rates, matching our results. Moreover, we included an updated data until 2017, that revealed the increasing trend of Black population SUD-related death rates during the last stage of the epidemic from 2015 to 2017. These findings suggest that decreasing physical distress by including preventive measures such as strategies to decrease morbidity of chronic conditions such as cardiovascular diseases, cancer, diabetes, and stroke may help lower SUD when used in conjunction with traditional approaches to prevent or treat SUD [39, 41].

The geographical patterns of the SUD-related mortality observed in our study revealed a series of spatially clustered sub-epidemics with different characteristics within the country. We found that areas in the Midwest surrounding the tri-state border of Ohio, Kentucky, and West Virginia had the highest RR of SUD-related mortality at national level. Counties within this hotspot had a risk of SUD-related death between 2.5 to 5.6 times higher compared to the rest of the country. Other areas with a significant spatial concentration of SUD-related deaths were found among the southern Pacific and mountain divisions in California, Nevada, Utah, Colorado, and New Mexico. The characteristics of the concentration of SUD-related deaths in these areas differ from the above-mentioned synthetic opioid sub-epidemic occurring in the Midwest. These differences included the substances driving the sub-epidemics as well as the temporal trending on SUD-related deaths (Southern Pacific trending decreasing while the Midwest is increasing). The spatiotemporal pattern of the RR of SUD-related deaths suggests a spread of the epidemic from Southwest to Northeast during the period of the study. This progression of the overdose mortality rates is attributed mainly to the interplay between illegal drugs coming from the southern boarders and prescription and synthetic opioids throughout the Midwest and Northeast States [40]. While the epidemic in the Southern Pacific division was fueled by methamphetamines with a substantial amount of heroin overdoses in New Mexico from 2013 onwards, the Northeast region showed a significant increase in the RR of SUD-related deaths and like in the Midwest, this sub-epidemic is led by prescription and synthetic opioids [7]. Both Southwestern and Northeastern areas reported high levels of physical and mental distress, which resulted positive associated to high risk of death by substance overdose in our analyses.

Our study had several limitations worth noting. The main limitation comes from the nature of the data, which relies in the autopsies' ability to detect and classify substances and circumstances causing the death. Firstly, we used deaths classified as unintentional substance poisoning (ICD-10 codes: X40, X41, X42, X43, X44) to estimate SUD mortality rates, assuming that

this classification is a proxy for the mortality rates of the SUD epidemic. This assumption excludes death counts from the overdoses with no information of the self-awareness of harm (IDC-10 codes from Y10 to Y14), and deaths by intentional sef-harm/suicide by substance overdose (IDC-10 codes from X60 to X64), which can be difficult to classify in practice. In addition, drugs causing the overdoses are difficult to categorize, and approximately 20% of the overdose death certificates do not include the involved substance [42]. Even when a drug is listed, a significant number of opioid-related poisonings were classified into the broader categories of other opioids (T40.2) or other and unspecified narcotics (T40.6). Multiple opioids deaths (which were the leading cause of deaths during the last periods) and opioids combined with other drugs were often involved in overdose incidences which did not identify the substance responsible for the overdose. Additionally, autopsies and death certificates can change among states, and our analysis did not take into consideration this variation in the classification for SUD-related mortality. Further efforts are needed to improve the quality of the characterizations of SUD-related deaths, and to standardize substance classification across states, as for example the inclusion of fentanyl into the ICD-10 codes.

Another important limitation is the self-reported nature of the physical and mental distress data, which could produce correlation among covariates, and some bias in our estimations [43]. The selection of our metrics was based on previous studies about the drivers of the addiction diseases, and the availability of the information at national level. Moreover, the BRFSS is designed to provide confident data about the mental and physical distress, and it is widely used by several studies because it includes two important independent health characteristics of the population [9, 44]. Finally, the last limitation is related to our analysis limited to 2017 due to the official source of data for mortality rates is provided always two years behind the current date, which corresponds to the data request process to the CDC which was conducted in 2019.

Despite these limitations, our study is one of the first to conduct a multilevel spatial characterization of the key individual and community-level drivers of the SUD-related mortality in the U.S. Collectively, our results suggest that individual and community-level risk factors are unevenly distributed across different demographic groups, generating a series of sub-epidemics emerging at different times and locations within the country. Moreover, the epidemic has been fueled by the introduction of different substances at different times, impacting the SUD-related mortality rate at different phases of the epidemic. Federal, state, and local governments in the U.S. have implemented multiple intervention measures to decrease SUD-related mortality rates such as restrictions on the prescribing of opioids, efforts to restrict the flow of illicit opioids, and enhancing access to naloxone. Although these efforts, among others, have been relatively successful in decreasing overdose mortality rates in general, the identification of the vulnerable populations and areas that contain the multiple sub-epidemics would enhance the ability to design prevention campaigns, which have proven more effective in managing other diseases than intervention approaches alone [11]. Aligned with the "Know your epidemic, know your response" approach, the detailed spatial and epidemiological description of the vulnerable populations at high risk of SUD-related mortality in the U.S generated in this study can be used to create targeted prevention strategies and to localize intervention campaigns. Microtargeting strategies based on the understanding of the spatial structure and the multifactorial nature of the addiction epidemic would facilitate the design of targeted integrated preventive therapies for early identification of diagnosis in the young adult population [6, 45].

## Supporting information

**S1 Text. Supplementary methods and results.**
(DOCX)

**S1 Fig. Discrimination analysis per substance reported as complementary cause of death.**
(A) Total population, (B) White males (C), and Black males.
(JPG)

**S2 Fig. Detailed demographics of the growth trends.** (Log10 (# Deaths by Substance Use Disorders / # Total Deaths)) of the substance use disorder (SUD) epidemic in the U.S. (2005–2017) by semester.
(JPEG)

**S3 Fig. Community-level exposure risk factors for the substance use disorder (SUD)-related mortality.** (A) Percentage of children living under poverty, (B) percentage of the adult population that consume alcohol excessively, (C) percentage of the uninsured population, (D) percentage of the adult population that consumes tobacco, (E) average of mentally unhealthy days, (F) average of physically unhealthy days. All values are averaged from 2010–2017 by county.
(JPEG)

**S1 Table. Demographic and socioeconomic characteristics of the stratified sample aggregated by deaths by substance use disorders (SUD) and other causes, 2005 to 2017.**
(DOCX)

**S2 Table. Complementary results of the generalized additive model association analysis for the county level covariates with the estimated degrees of freedom results for multilevel and factor smooth interaction models.**
(DOCX)

**S3 Table. Identified clusters of deaths by substance use disorders from the U.S. individual mortality, 2005 to 2017.**
(DOCX)

## Acknowledgments

The authors thank the U.S. Center for Disease Control and Prevention (CDC) for collecting and releasing the data used in this study.

## Author Contributions

**Conceptualization:** Andrés Hernández, Minxuan Lan, Neil J. MacKinnon, Adam J. Branscum, Diego F. Cuadros.

**Data curation:** Andrés Hernández, Minxuan Lan.

**Formal analysis:** Andrés Hernández, Minxuan Lan, Diego F. Cuadros.

**Investigation:** Andrés Hernández, Minxuan Lan, Neil J. MacKinnon, Diego F. Cuadros.

**Methodology:** Andrés Hernández, Minxuan Lan, Adam J. Branscum, Diego F. Cuadros.

**Supervision:** Neil J. MacKinnon, Diego F. Cuadros.

**Validation:** Andrés Hernández.

**Writing – original draft:** Andrés Hernández, Diego F. Cuadros.

**Writing – review & editing:** Minxuan Lan, Neil J. MacKinnon, Adam J. Branscum.

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
