## [Decision Letter · Decision Letter 0]

1 Feb 2021

PONE-D-20-31970

“Know your epidemic, know your response”: Epidemiological assessment of the substance use disorder crisis in the United States

PLOS ONE

Dear Dr. Cuadros,

Thank you for submitting your manuscript to PLOS ONE. After careful consideration, we feel that it has merit but does not fully meet PLOS ONE’s publication criteria as it currently stands. Therefore, we invite you to submit a revised version of the manuscript that addresses the points raised during the review process. 

We look forward to receiving your revised manuscript.

Kind regards,

Arsham Alamian, PhD, MSc, FACE

Academic Editor

PLOS ONE

Journal Requirements:

2) You indicated that ethical approval was not necessary for your study. We understand that the framework for ethical oversight requirements for studies of this type may differ depending on the setting and we would appreciate some further clarification regarding your research. Could you please provide further details on why your study is exempt from the need for approval and confirmation from your institutional review board or research ethics committee (e.g., in the form of a letter or email correspondence) that ethics review was not necessary for this study? Please include a copy of the correspondence as an "Other" file.

3) Please include captions for your Supporting Information files at the end of your manuscript, and update any in-text citations to match accordingly. Please see our Supporting Information guidelines for more information: http://journals.plos.org/plosone/s/supporting-information.

Reviewers' comments:

Reviewer's Responses to Questions

**Comments to the Author**

1. Is the manuscript technically sound, and do the data support the conclusions?

Reviewer #1: Yes

Reviewer #2: Yes

2. Has the statistical analysis been performed appropriately and rigorously? 

Reviewer #1: Yes

Reviewer #2: Yes

3. Have the authors made all data underlying the findings in their manuscript fully available?

Reviewer #1: Yes

Reviewer #2: Yes

4. Is the manuscript presented in an intelligible fashion and written in standard English?

Reviewer #1: Yes

Reviewer #2: Yes

5. Review Comments to the Author

Reviewer #1: “Know your epidemic, know your response”: Epidemiological assessment of the substance use disorder crisis in the United States

This manuscript describes the substance use disorder epidemic that is currently occurring in the United States. Although SUD is occurring at very high rates, not enough information is known about the demographics or socioeconomic factors related to individuals using substances. This study used data obtained from the CDC from 2005-2017 to examine characteristics of SUD. Findings were that White males consistently had highest SUD mortality rate from 2005-2017. There was a surge of increase in mortality in Black males from 2014-2017. SUD clustering showed there were more clusters in the west and Midwest. The researchers also found that having mental health distress or conditions significantly increased the relative risk of acquiring SUD. Overall, this is a very interesting manuscript that contributes to the scientific literature. However, the writing could be more clear at times. Suggestions for improvement are provided below:

Abstract:

1. For this sentence in the conclusion “we found that this sad epidemic in the U.S. is characterized by…” I got confused with “sad” standing for substance abuse disorder – perhaps change to SUD or rephrase

2. For the first sentence in the results section, consider spelling out “U.S.” to United States so it is more clear that the next sentence begins with “white”, otherwise this appears to be a run on sentence.

Introduction:

1. Regarding the definition of SUD, the manuscript states that the person that engages in the behavior has knowledge of the harmful consequences of engaging in excessive substance use. However, how is this measured and how do we know the individual is aware of the harm? In the research methods section the data collected is about substance use and behavior, not knowledge of the behavior.

2. End of page 3, states that “know your epidemic, know your response” framework shifted from coercive strategies to targeted prevention strategies. What does this mean? How were they coercive before, how did the shift occur to targeted prevention, and clarify what this means?

Methods:

1. Page 5, second to last paragraph, last sentence should be “publicly” (not publicity)

2. The statistical methods seem to be adequately explained and appropriately applied to the data set.

Results:

1. Page 7, paragraph 1 – for the sentence that says mortality in Black males has risen sharply since 2014, 40.00 SUD-related deaths per 1,000 total deaths is listed, however I would like to see a comparison number for white males.

2. Page 7 – table 1, figure 1 and figure 2 use the terminology substance abuse disorder (SAD) while the rest of the manuscript has been using substance use disorder (SUD) – clarification is needed here

3. Page 8, paragraph 1 – states that there is no statistical difference in population odds of SUD-related death for males and females, this needs clarification as males suffer from disproportionately high SUD and have shown highest mortality 2005-2017 stated multiple places elsewhere.

Discussion:

1. Middle paragraph on page 11 – the description of the epidemic in the southern Pacific/mountain region is a little unclear to me. 2-3 more sentences of elaboration would be helpful to contrast with the epidemic occurring in the Midwest and other regions.

2. Appropriate limitations are mentioned.

Figures:

1. Figure 2 – description of the figure and short analysis contrasting A and B would be helpful/useful

2. Figure 4 – B – clarify ‘intentional’ substance use disorders

Recommendation: Accept subject to revision

Reviewer #2: Re: PONE-D-20-31970

Diego Cuadros

“Know your epidemic, know your response”: Epidemiological assessment of the substance use disorder crisis in the United States

1. Topic of considerable interest given Case and Deaton’s PNAS paper and book on “Deaths of Despair” and the noted increased mortality due to suicide, overdose, accidents in select groups in the last 5-8 years.

2. The authors identify areas in the U.S. with increase deaths by SUD using CDC data 2005-2017.

3. Methodology excellent

4. Results

Increase of death due to SUD

Males

White with a sharp increase in Blacks since 2014

Increase 25-29, higher education (Did they look at college/no college, single, divorced, poverty, physical illness?)

Certain States, Mid-West followed by Southern Pacific, Mountain States, and North East States showed increase.

The opportunity to check out the Case and Deaton findings on education is partially lost by the absence of 2.18% of the data on education. The figures are presented for primary/secondary/college level. Can the authors look further at college graduate vs. others?

6. PLOS authors have the option to publish the peer review history of their article (what does this mean?). If published, this will include your full peer review and any attached files.

Reviewer #1: No

Reviewer #2: No

---

## [Author Response · Author response to Decision Letter 0]

23 Feb 2021

We appreciate the opportunity to resubmit our work to the journal. We would like to thank the editor and the reviewers for assessing our work and for their valuable feedback and suggestions that have improved our manuscript. Please find below a point-by-point reply that addresses each of the journal and the reviewers’ comments. We have also incorporated these suggestions in the revised manuscript as noted below. We would be pleased to address any further points that the editor or reviewers may find unsatisfactory.

JOURNAL COMMENTS:

Style requirements were followed in the revised version of the manuscript according to the guidelines provided by the journal, including the file naming patterns.

2. You indicated that ethical approval was not necessary for your study. We understand that the framework for ethical oversight requirements for studies of this type may differ depending on the setting and we would appreciate some further clarification regarding your research. Could you please provide further details on why your study is exempt from the need for approval and confirmation from your institutional review board or research ethics committee (e.g., in the form of a letter or email correspondence) that ethics review was not necessary for this study? Please include a copy of the correspondence as an "Other" file.

As we stated in our manuscript, we used for our analyzes the publicly available vital statistics micro-data from the Centers for Disease Control (CDC) compiled by the National Center for Health Statistics (NCHS) (https://www.cdc.gov/nchs/data_access/vitalstatsonline.htm). These data are neither identifiable nor private and thus do not meet the federal definition of “human subject” as defined in 45 CFR 46.102. Therefore, these research projects do not need to be reviewed and approved by an Institutional Review Board, (IRB). For your reference, a similar study using the same data sources as our study and published last year in the journal also stated that because their study involved analysis of existing, deidentified data, it was exempt from human subjects review (“Glei, Dana A., and Samuel H. Preston. "Estimating the impact of drug use on US mortality, 1999-2016." PloS one 15.1 (2020): e0226732.) 

3. Please include captions for your Supporting Information files at the end of your manuscript, and update any in-text citations to match accordingly.

The supporting information captions were included at the end of the manuscript according to the style of the reference provided by the editor. (Page 20, Line 1, Supporting Information Section)

REVIEWER COMMENTS:

REVIEWER 1

Abstract:

1. For this sentence in the conclusion “we found that this sad epidemic in the U.S. is characterized by…” I got confused with “sad” standing for substance abuse disorder – perhaps change to SUD or rephrase

We appreciate the feedback of the reviewer. The term Substance Use Disorder (SUD) replaced all other related abbreviations along the manuscript for consistency, accordingly to the reviewer suggestion. (Page 2, line 3, Abstract section), (Page 2, line 16, Abstract section), (Page 8, line 16, Table 1 caption), (Page 8, line 18, Fig 1 caption), (Page 8, line 20, Fig 2 caption), (Page 10, line 4, Table 2 caption), (Page 10, line 7, Fig 3 caption), (Page 12, line 5, Fig 4 caption), (Page 13, line 2, Table 3 caption), (Page 12, line 5, Discussion section).

2. For the first sentence in the results section, consider spelling out “U.S.” to United States so it is clearer that the next sentence begins with “white”, otherwise this appears to be a run on sentence.

The suggestion was included in the reviewed version of the manuscript. (Page 2, line 11, Abstract section).

Introduction:

1. Regarding the definition of SUD, the manuscript states that the person that engages in the behavior has knowledge of the harmful consequences of engaging in excessive substance use. However, how is this measured and how do we know the individual is aware of the harm? In the research methods section the data collected is about substance use and behavior, not knowledge of the behavior.

We appreciate the feedback from the reviewer, and we agree with the reviewer about the need of further clarification of the definition of Substance Use Disorders (SUD) mortality. Our research methods focus on the analysis of unintentional (accidental) deaths by substance overdose as defined by the ICD10 standard for causes of death codes X40, X41, X42, X43, X44. The other causes of deaths including codes X60, X61, X62, X63, X64 (Deaths by Intentional Self- harm/Suicide caused by substance overdose), and deaths with no information about the self-intention (codes Y10, Y11, Y12, Y13, Y14) were used as control events (outcome ‘0’) in our binary outcome. This choice made the assumption that death by accidental substance overdose is a proxy for the mortality of substance abuse disorders that does not include the intention of self-harm. This choice was based on the literature review and implies some limitations worth noting. As a result of the reviewer suggestion, we included in the revised version of the manuscript a mention to this limitation as the data source does not specify previous medical history of deceased individuals. (Page 14, line 14, Discussion section, limitations).

2. End of page 3, states that “know your epidemic, know your response” framework shifted from coercive strategies to targeted prevention strategies. What does this mean? How were they coercive before, how did the shift occur to targeted prevention, and clarify what this means?

“Know your epidemic, know your response” is a framework that prioritize preventive approaches (e.g., education campaigns, targeting cause of epidemics) more than reactive measures (e.g., vaccination, medical treatment) for mitigating epidemics. In principle, this framework has been proposed to target HIV, however several studies reported that components of this approach are also applicable to other epidemics. Additionally, this framework states that the individual’s knowledge about risk is critical for the ability to respond epidemics with risk reduction strategies, and the individual social behaviors are determinants on the prevalence of the epidemics. As a suggestion of the reviewer’s feedback, we include an updated bibliography highlighting the aforementioned elements, and rephrasing the concept of coercive strategies for clarifying in the reviewed version of the manuscript. (Page 4, line 4, Introduction section).

Methods:

1. Page 5, second to last paragraph, last sentence should be “publicly” (not publicity)

We appreciate the careful examination of the reviewer. We included the suggestion in the reviewed version of the manuscript. (Page 5, line 20, Methods, Data source Description section).

Results:

1. Page 7, paragraph 1 – for the sentence that says mortality in Black males has risen sharply since 2014, 40.00 SUD-related deaths per 1,000 total deaths is listed, however I would like to see a comparison number for white males.

As a result of the reviewer’s suggestion. We included in the revised version of the manuscript the comparison of the temporal trend of SUD-related deaths of black and white males. (Page 7, line 11, Results section). 

2. Page 7 – table 1, figure 1 and figure 2 use the terminology substance abuse disorder (SAD) while the rest of the manuscript has been using substance use disorder (SUD) – clarification is needed here

We appreciate the feedback of the reviewer. The term Substance Use Disorder (SUD) was included along all the manuscript accordingly to the reviewer suggestion. (Page 2, line 3, Abstract section), (Page 2, line 16, Abstract section), (Page 8, line 16, Table 1 caption), (Page 8, line 18, Fig 1 caption), (Page 8, line 20, Fig 2 caption), (Page 10, line 4, Table 2 caption), (Page 10, line 7, Fig 3 caption), (Page 12, line 5, Fig 4 caption), (Page 13, line 2, Table 3 caption), (Page 12, line 5, Discussion section).

3. Page 8, paragraph 1 – states that there is no statistical difference in population odds of SUD-related death for males and females, this needs clarification as males suffer from disproportionately high SUD and have shown highest mortality 2005-2017 stated multiple places elsewhere.

Taking into account the suggestion of the reviewer, we added in the discussion section further mention about this result, and we included new bibliography that analyzed this topic with similar results as ours. Even though we agree with the reviewer that men suffer a disproportionately high SUD-related deaths, our association analysis resulted in a non-significant effect of gender on the risk of death by SUD likely caused by the relationship between gender, morbidity and other socioeconomic factors included in our adjusted model. (Page 13, line 14, Discussion section).

Discussion:

1. Middle paragraph on page 11 – the description of the epidemic in the southern Pacific/mountain region is a little unclear to me. 2-3 more sentences of elaboration would be helpful to contrast with the epidemic occurring in the Midwest and other regions.

As a result of the suggestion of the reviewer, we add more details and updated bibliography regarding the differences of the epidemic occurring in the Midwest and North East compared to the southern Pacific/mountain regions. (Page 14, line 2, Discussion section).

Figures:

1. Figure 2 – description of the figure and short analysis contrasting A and B would be helpful/useful

We added further comparison among the rates between black and white male population in Figure 2. We also rounded the morality rates to 2 decimals to compute the percentage of increasing rates for black and white males from 2014 to 2015. (Page 7, line 6, Abstract section).

2. Figure 4 – B – clarify ‘intentional’ substance use disorders

We appreciate the feedback of the reviewer and we agree with the need of clarification about the use of ‘intentional’ and ‘unintentional’ in the context of death by substance overdose. Our research methods include unintentional accidental deaths by substance overdose as defined by the ICD10 codes for causes of death X40, X41, X42, X43, X44. The other causes of deaths including codes X60, X61, X62, X63, X64 (Deaths by Intentional Self- harm/Suicide caused by substance overdose) and Y10, Y11, Y12, Y13, Y14 (Undetermined intent substance overdose death) were used as control cases in our binary outcome. This clarification is included in the reviewed version of the manuscript. (Page 14, line 14, Discussion section).

REVIEWER 2

1. Topic of considerable interest given Case and Deaton’s PNAS paper and book on “Deaths of Despair” and the noted increased mortality due to suicide, overdose, accidents in select groups in the last 5-8 years.

We appreciate the interesting insights of the reviewer. We also agree that the reference the author is citing is an important piece of the literature related to our study, given the fact they used the same data source for the death counts (CDC Wonder micro-files), and validated some of our findings. As a result, we added this reference to our bibliographic review in the revised version of our manuscript and included more details to the discussion based on the findings of Case and Deaton’s work. (Page 19, line 14, Reference section).

4. Results

Increase 25-29, higher education (Did they look at college/no college, single, divorced, poverty, physical illness? The opportunity to check out the Case and Deaton findings on education is partially lost by the absence of 2.18% of the data on education. The figures are presented for primary/secondary/college level. Can the authors look further at college graduate vs. others?

Our study reported the results from a statistical association analysis between the individual risk of death caused by unintentional substance poisoning and several demographic and socio-economic covariates, including (age group, gender, education, socio economic status and marital status), as well as self-reported days of physical and mental distress from the Behavioral Risk Factor Surveillance System averaged for the time period of 2010 – 2017 and aggregated at the county level.

From the results of our association analysis, we agree with the reviewer that Case and Deaton’s research is an important piece of the literature to be discussed within the context of our study. However, Case and Deaton study shows several differences and similarities with our study that are worth noting because they limit our ability to compare both findings of both studies, especially in the relationship of educational level and SUD-related death risk. As a result of the reviewer’s suggestion, we summarized the similarities and differences of our study with the Case and Deaton’s work and included them in the revised version of our manuscript. (Page 13, Lines 10, Discussion section)

Among similarities and differences, Case and Deaton’s research offers a wider range of death causes than the focus of our study. They also concentrated a significant portion of their study on midlife adults, and highlighted suicides, alcohol-related and overdose as one of the critical drivers of the mortality rates increase for whites non-hispanic (WNH) in the United States. In addition, they found that educational level was critical for several aspects of the death rate increment from 1999 - 2015. In our study, we excluded intentional self-harm, and other substances apart from (Heroin, Methadone, Other opioids, synthetic narcotics, cocaine, unspecified narcotics) and included a wider range of age groups (from 5 to 84 years). As age is related to educational level, this could explain that college education did not show statistical significance, but high school resulted in an associated factor for the increased risk of SUD-related death.

Another difference is the time range of both studies. While Case and Deaton’s focused from 1999 to 2015, our study was conducted from 2005 to 2017. This is especially important because Case and Deaton study reported an increase of WNH death rates from 1999 to 2015, and a parallel decrease of death rates for Blacks and Hispanic population on the same period. Our study shows that Black SUD-related death rates rised sharply from 2013 and reach the WNH SUD-related death rates in 2017, period which was out of the scope of the Case and Deaton’s study. 

Finally, Case and Deaton were similar to our methods in the use of the same data sources for the computation of mortality rates, and a resulting increase of morbidity of physical and mental health linked to the increase of death rates. This offers an interesting insight to our study because they exposed the lack of job opportunities and education as a potential cause of the increasing death rates. In addition, they validated our spatiotemporal pattern which suggest a diffusion of the epidemic from the Southwest to the North Eastern and Mideastern states and an unclear association between gender and wealth with an increasing of alcohol, suicide and overdose related death rates that is a super set of the SUD-related mortality rates of our study.

---

## [Decision Letter · Decision Letter 1]

28 Apr 2021

“Know your epidemic, know your response”: Epidemiological assessment of the substance use disorder crisis in the United States

PONE-D-20-31970R1

Dear Dr. Cuadros,

We’re pleased to inform you that your manuscript has been judged scientifically suitable for publication and will be formally accepted for publication once it meets all outstanding technical requirements.

Kind regards,

Arsham Alamian, PhD, MSc, FACE

Academic Editor

PLOS ONE

Additional Editor Comments (optional):

Reviewers' comments:

Reviewer's Responses to Questions

**Comments to the Author**

1. If the authors have adequately addressed your comments raised in a previous round of review and you feel that this manuscript is now acceptable for publication, you may indicate that here to bypass the “Comments to the Author” section, enter your conflict of interest statement in the “Confidential to Editor” section, and submit your "Accept" recommendation.

Reviewer #1: All comments have been addressed

2. Is the manuscript technically sound, and do the data support the conclusions?

Reviewer #1: Yes

3. Has the statistical analysis been performed appropriately and rigorously? 

Reviewer #1: Yes

4. Have the authors made all data underlying the findings in their manuscript fully available?

Reviewer #1: Yes

5. Is the manuscript presented in an intelligible fashion and written in standard English?

Reviewer #1: Yes

6. Review Comments to the Author

Reviewer #1: This version of the manuscript is much better. I believe both my comments and the other reviewer's comments appear to be addressed. Good detail that elaborates on the differing epidemics in South vs Midwest, helpful clarification was provided regarding statistical analysis as well. Overall good job. I feel that this manuscript needs no further revision at this point.

7. PLOS authors have the option to publish the peer review history of their article (what does this mean?). If published, this will include your full peer review and any attached files.

Reviewer #1: **Yes: **Kathryn Gerber

---

## [Editor Report · Acceptance letter]

3 May 2021

PONE-D-20-31970R1 

“Know your epidemic, know your response”: Epidemiological assessment of the substance use disorder crisis in the United States 

Dear Dr. Cuadros:

I'm pleased to inform you that your manuscript has been deemed suitable for publication in PLOS ONE. Congratulations! Your manuscript is now with our production department. 

Kind regards, 

on behalf of

Dr. Arsham Alamian 

Academic Editor

PLOS ONE